# A shift in transitional forests of the North American boreal will persist through 2100
Paul M. Montesano [1,2] ✉, Melanie Frost [1,3], Jian Li[1,3], Mark Carroll[1], Christopher S. R. Neigh [1], Matthew J. Macander [4], Joseph O. Sexton[5] & Gerald V. Frost [4]

High northern latitude changes with Arctic amplification across a latitudinal forest gradient suggest a shift towards an increased presence of trees and shrubs. The persistence of change may depend on the future scenarios of climate and on the current state, and site history, of forest structure. Here, we explore the persistence of a gradient-based shift in the boreal by connecting current forest patterns to recent tree cover trends and future modeled estimates of canopy height through 2100. Results show variation in the predicted potential height changes across the structural gradient from the boreal forest through the taiga-tundra ecotone. Positive potential changes in height are concentrated in transitional forests, where recent positive changes in cover prevail, while potential change in boreal forest is highly variable. Results are consistent across climate scenarios, revealing a persistent biome shift through 2100 in North America concentrated in transitional landscapes regardless of climate scenario.

Recent shifts (the growth, dieback, mortality, and composition change of vegetation) in boreal forest structure and density[1–4] suggest a high northern latitude vegetation domain in flux[5,6]. This domain is experiencing changes to patterns of vegetation, permafrost, and disturbance regimes that suggest a shift towards an increased presence of trees and shrubs[7–14]. This domain harbors important global carbon sinks, supports wildlife habitat, and is associated with subsistence activities that have supported the persistence of boreal landscapes as humans have come to know, interact with, and benefit from them.

The shifts are occurring across a latitudinal gradient of forest structure. This gradient coincides with broad-scale climatic patterns[15], and changes to the boreal forest within the gradient are dependent in part upon the interaction of precipitation and temperature[16,17], as well as with changes in wildfire intensity[18,19]. Shifts within this gradient may be particularly conspicuous and consequential in the transitional landscapes of the boreal forest (taiga) - tundra ecotone[20–22], where the increased presence, cover, and height of woody vegetation across broad extents may affect regional albedo[23,24]. As the high northern latitudes continue to experience disproportionate increases in temperature, variability in precipitation, extreme events, and a general departure from historical climatic expectations and disturbance regimes associated with Arctic amplification[25,26], shifts in vegetation may persist.

The magnitude and direction of these shifts, and the novel spatial configurations in vegetation that they produce, can assume a variety of forms that vary in their ability to persist through time[27]. These forms include woody (trees and shrubs) structure domain expansion/contraction,

densification/diffusification, any combination of associated productivity[28] and clonal group dynamics[29,30], and can vary according to site-scale and regional environmental characteristics[31–33]. They will have an impact on global climate, biodiversity, and human societies in and near the boreal domain.

Studies across the circumboreal region reveal mixed responses of high northern latitude forest growth in response to change in temperature in the Arctic and Sub-Arctic[34–40]. Evidence of shifting boreal vegetation comes from satellite interpretations of recent vegetation 'greening' and 'browning'[41–43], observations of vegetation height change[44], and estimates of increases in net primary production[45] and reduced thermal constraints to growth[46]. Methods for predicting boreal forest growth based on future climate have been explored[47], as changes in height growth through time indicate a growth response to environmental conditions[33] and a key axis of environmental variation[48]. The persistence of these boreal shifts may depend partly on the future scenarios of anthropogenic drivers of climate change[49,50], but also on the current state, and recent site history, of forest structure.

Here we adopt an approach using current patterns in the gradient of forest structure (Supplementary Fig. 1) to connect site history of recent tree cover trends with future potential for canopy height change. To do this, we characterize landscape patterns, using hydrological basins to classify mesoscale spatial extents according to the spatial pattern of forest structure[22,51–56] in boreal forests and transitional forests of the taiga-tundra ecotone (TTE). We then use the large archive of current observations of both boreal vegetation height from the Ice, Cloud, and Elevation Satellite-2 (ICESat-2) lidar, which has been recording contemporary vegetation

[1]NASA Goddard Space Flight Center, Greenbelt, MD, USA. [2]ADNET Systems, Inc., Bethesda, MD, USA. [3]ASRC Federal InuTeq, Beltsville, MD, USA. [4]Alaska Biological Research, Inc., Fairbanks, AK, USA. [5]TerraPulse, Inc., Potomac, MD, USA. ✉e-mail: paul.m.montesano@nasa.gov

canopy heights since 2018[57] coupled with environmental covariates of current gridded bioclimatic, permafrost, and soil data to model current and predict future patterns of boreal vegetation height based on downscaled future bioclimatic predictions derived from climate scenarios from the Coupled Model Intercomparison Project 6 (CMIP6) Shared Socioeconomic Pathways (SSPs). Using current landscape patterns describing the forest gradient, we quantify the variation in the change of these future height predictions and associate them with site histories of recent structural changes in boreal tree canopy cover to evaluate the persistence through 2100 of the ongoing biome shift. This assessment of future predictions of potential growth with recent structure trends based on current structure patterns offers insight into the variation in the fate of the heterogeneous boreal forest across North America.

## Results

### Landscape forest gradient classification
For the study domain (Supplementary Fig. 2) we assembled a set of 16,559 hydrobasins (landscapes) that were classified according to the prevailing spatial pattern of their current forest gradient in Landsat-derived tree canopy cover (Fig. 1). These landscapes were used to connect forest structure changes predicted for the future with those observed from recent satellite-based trends. The inset histogram and the median area of these landscapes (359 km²) clarifies the mesoscale level of assessment used to establish connection between future and recent forest structure dynamics.

### Bioclimatic predictions of current canopy height
We trained, tested, and applied a bioclimatic prediction model to map coarse-scale (2.5 degree) current potential forest canopy height. This model related a suite of current coarse-scale gridded (≥250 m) environmental covariate predictors describing climate and soil characteristics with lidar-derived observations of current canopy height from the ATL08 Vegetation Height product from ICESat-2[58]. It was trained with 80% of gridded observations of height (n = 1,082,746) built from filtered ICESat-2 ATL08

observations of the 98th percentile of canopy height (n = 19,875,592, Supplementary Fig. 3), and applied to the suite of current predictors (c. 2020) across the *prediction* domain (Supplementary Fig. 2) of the North American boreal, with 20% of the observations reserved as a model test set. In comparison to reference gridded height, the mapped predictions of this test set explained most of the variability with an R² value of 0.54 and a positive bias of 2.75 m (p < 0.001) (Supplementary Fig. 6).

We evaluated the distributions of these canopy height predictions with [1] the portion of the model test set of the reference grid-based ATL08 canopy heights with which they spatially correspond, and [2] the entire set of point-based ATL08 canopy heights for the full set of predictions (Supplementary Fig. 7) as well as for each of the 6 landscape forest gradient classes (Supplementary Fig. 8). The performance of the model of current height predictions established a basis for understanding the differences (change) in this model's future climate-scenario based ensemble predictions of height from these current predictions.

The summaries for each landscape forest gradient class of the three distributions of current canopy heights (reference point-based ATL08, grid-based ATL08, and model predicted) (Supplementary Table 2) show each class's model predicted median canopy height was more similar to the grid-based ATL08 observations (≤0.7 m difference of medians) than to those of the point-based ATL08 (≤1.5 m difference of medians), This reflects the effect of the grid-based aggregation compared to the finer scaled (20 m length) individual point-based ATL08 observations. For the total study domain, the distribution of model predicted canopy heights feature a slightly higher median canopy height (6.2 m) and smaller variances (0.1–3.4 m) than the corresponding grid-based ATL08 results (5.8 m and 0.6−4.6 m, respectively) across landscape forest gradient classes, highlighting the general tendency of the model to predict fewer extremely short or tall canopy heights (Supplementary Fig. 7). This apparent decreased sensitivity to extreme values is further evident in light of the larger variances associated with the point-based ATL08. This decreased sensitivity in the bioclimatic prediction of canopy height is further reflected in the bias reported (Supplementary Fig. 5), however, this bias is mitigated by virtue of an analysis of

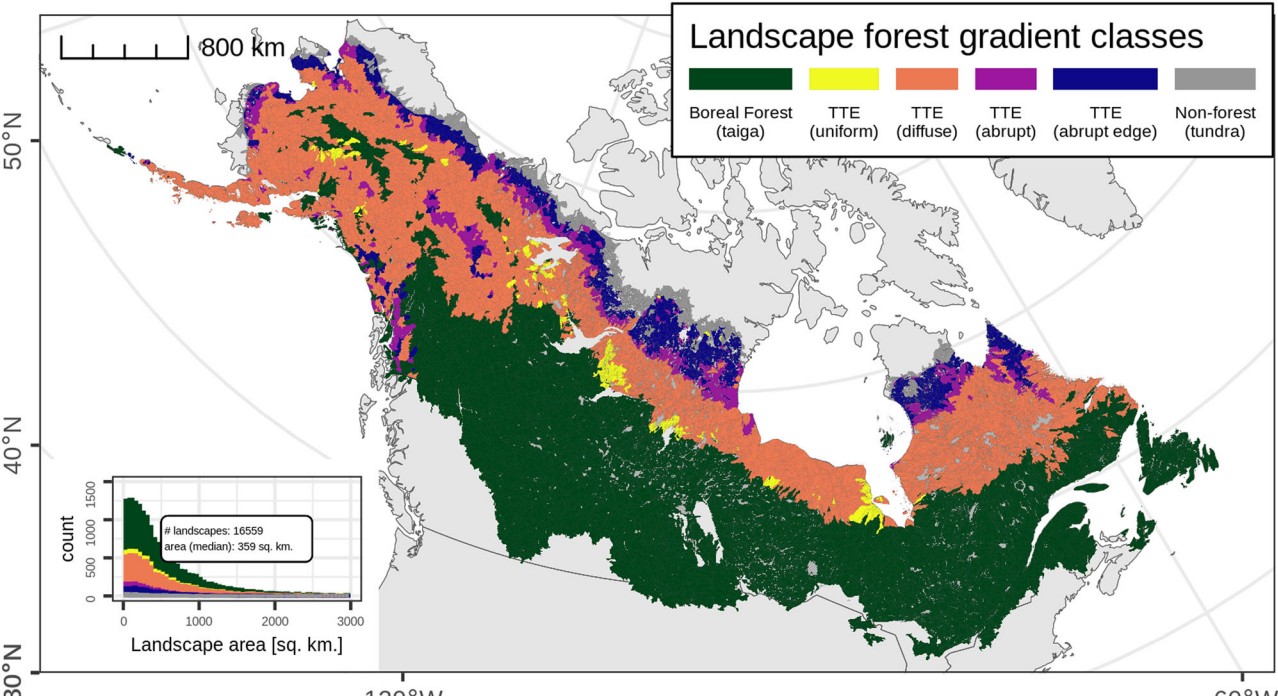

**Fig. 1 | Landscape forest gradient classes of the North American boreal.** Forest gradient classes shown for landscapes derived from level 8 hydrobasins intersecting the study domain (n = 16,559). These landscapes combine those associated with the taiga-tundra transition zone (TTE)[22], with those of the boreal forest and a small set associated with adjacent tundra.

the differences (change) between future predictions and current predictions, reported below.

## Ensemble predictions of potential canopy height change through 2100

We modeled future potential canopy height based on mapped bioclimatic predictions from 9 global climate models across four 20-year time intervals from 2021 to 2100 and 4 CMIP6 SSPs. The derived 134 individual potential canopy height predictions were compiled to 16 ensemble median value maps, one for each time interval and CMIP6 SSP. The gridded current canopy height prediction was subtracted from each of these 16 ensemble prediction scenarios to show predicted differences in canopy height from current conditions through time and across climate outcomes (Fig. 2). Across the North American boreal domain these ensembles show broad spatial variation in predicted canopy height differences depending on SSP emission scenario and time interval, yet show consistency in how vegetation structure responds across these scenarios.

Positive differences in height indicate net growth in forest structure at the landscape scale, while negative differences provide an indication of some combination of dieback and mortality. Notable positive differences in canopy height are predicted for transitional landscapes where forest cover becomes increasingly diffuse. This variation in the prediction of canopy height change corresponds with forest gradient class. Positive differences in canopy height are predicted for all transitional forest gradient classes, as well as the 'Non-forest (tundra)' class. A near 0 potential net change with broad variation in median canopy height across the 'Boreal Forest (taiga)' gradient class through 2100 suggests a highly variable future for vegetation structure within the current interior of the North American boreal forest (Fig. 3).

## Linking current structure, recent trends, and predictions of future change

We linked observations of recent history and predictions of future changes in forest structure using current patterns in its gradient across the North American boreal for 12,422 landscapes. Figure 4 shows observations of recent rates of change in tree canopy cover from 1984 to 2020, highlighting the strongest positive change towards the northern (cold) edge of the study domain. The largest positive (median) values are in the most abundant transitional forest class ('TTE (diffuse)'), which is second in total area to only the 'Boreal Forest (taiga)' class. We note geographic variation between recent trends in northern landscapes of central Canada, immediately west of Hudson Bay, and the rest of the northern portions of the domain. Median p-values < 0.1 (Supplementary Fig. 9) for most landscapes suggest a preponderance of significant trends at mesoscales.

Figure 5 summarizes the link between observations of recent trends and potential future change in the structure of the North American boreal through 2100. These results reflect continued climate warming based on a "Middle of the Road" narrative associated with a future in which trends in the use of fossil fuels and economic growth are similar to those of the recent past (an additional radiative forcing of 4.5 W m$^{-2}$; SSP245), and thus represent potential structural change expected if human development proceeds as it has[49,59]. The 9 classes summarize forest structure dynamics based on recent (tree canopy cover) and future (potential canopy height) change across boreal landscapes for which both change estimates were made. Forest structure changes are summarized into classes based on median values captured from individual landscapes as negative (decreasing; −), no change (stable; 0), or positive (increasing; +). Transitional forest landscapes (TTE) that feature both positive recent and potential future structural changes account for >1.31 × 10$^6$ sq. km. and account for ~20% of the domain, while the same combination of change classes across boreal

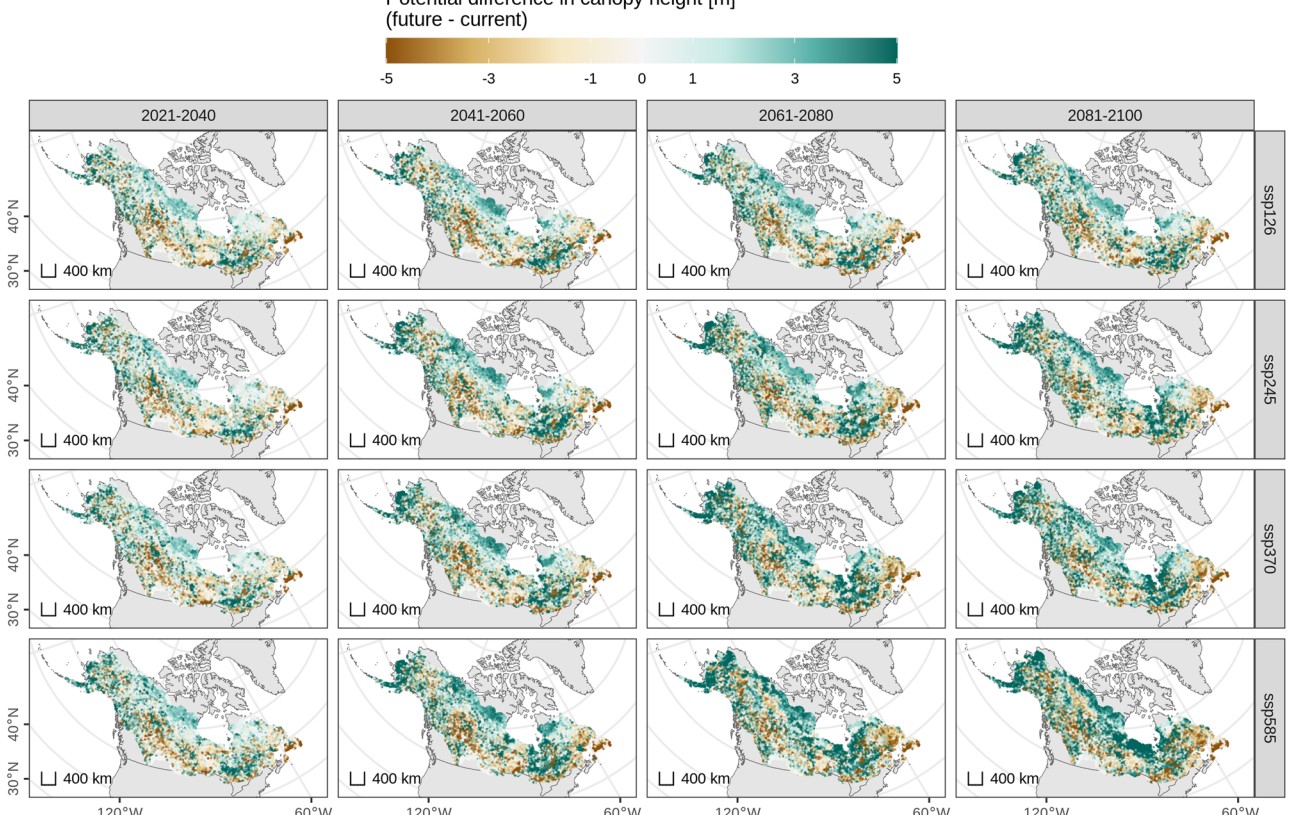

**Fig. 2 | Potential future differences in canopy height.** Spatial pattern of ensemble estimates showing the potential difference between future and current canopy height predictions across the 4 time periods and CMIP6 SSPs. Spatial patterns persist through time, intensify through 2100, and are consistent in direction across the range of climate scenarios. Positive potential height changes are concentrated in and near the northern portions of the domain.

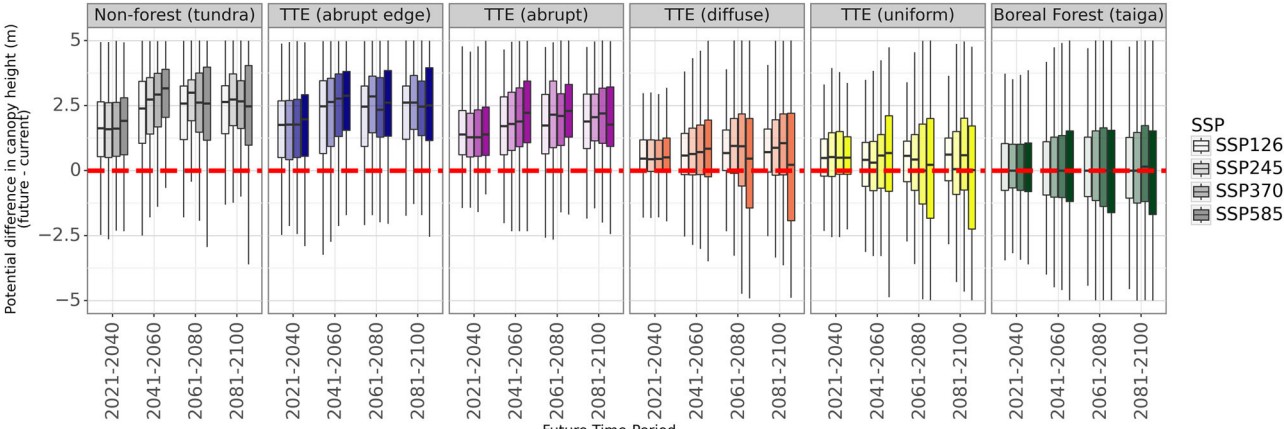

**Fig. 3 | Potential future differences in height by landscape class and climate scenario.** Summary for 6 forest gradient classes of ensemble estimates of the potential difference between future and current canopy height predictions across the 4 time periods and CMIP6 climate scenarios (shown with progressively darker shading representing the increasing intensity of emission scenarios). Potential canopy height changes across the boreal-tundra forest gradient are the most positive in landscapes that currently feature the least forest. The 4 boxplots associated with each time period vary in their shading (from left to right, lightest to darkest) in accordance with the severity of the 4 CMIP6 climate scenarios. Boxplots show the median (black line), the lower and upper hinges of the boxplots extend across the interquartile range (IQR), and the whiskers extend from each hinge through to 1.5 * IQR in either direction.

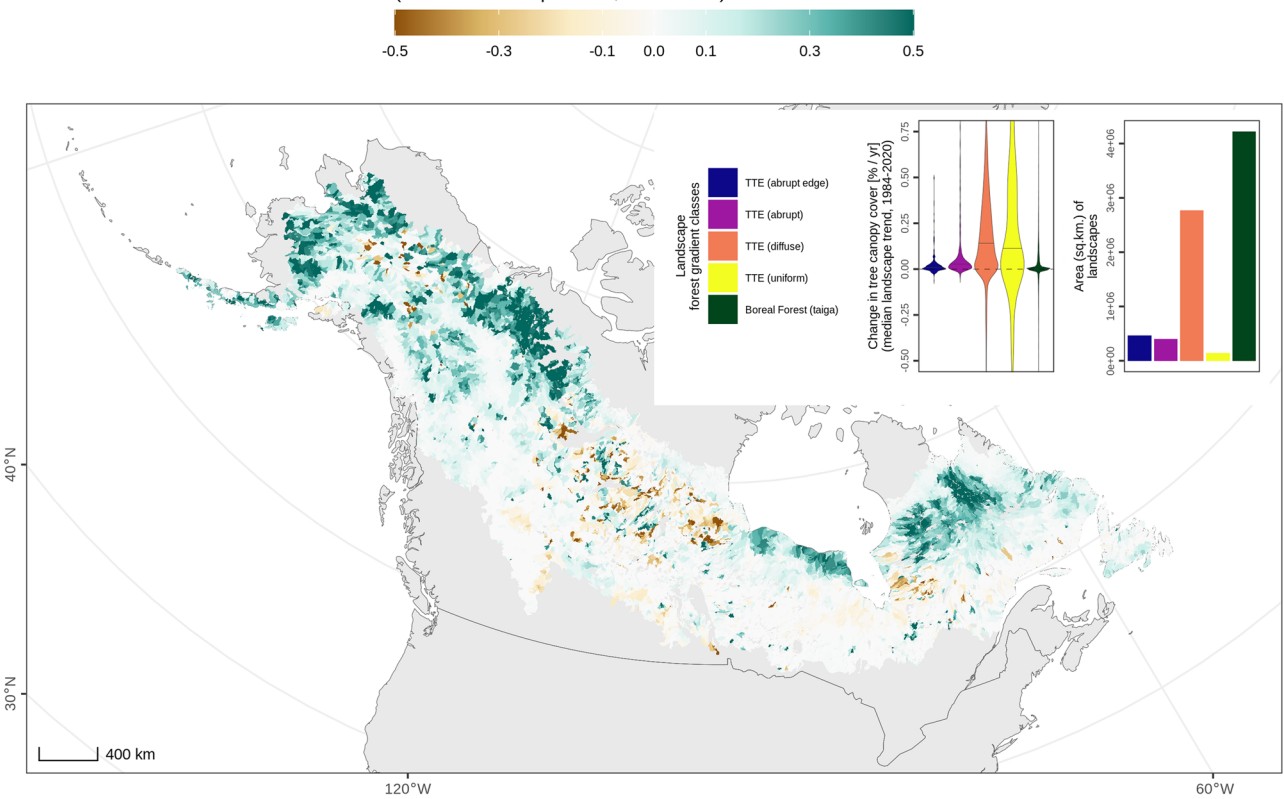

**Fig. 4 | Landscape changes in tree canopy cover in the North American boreal.** Recent observations from Landsat reveal patterns in tree canopy cover trends (1984–2020) for 12,422 landscapes. Landscapes classified according to forest gradient class[22] show positive median changes are strongest in transitional landscapes.

landscapes (>0.39 × 10⁶ sq. km.) account for ~6% of the domain. Supplementary Fig. 10 prevents summary results for the remaining 3 SSPs through 2100.

## Discussion

These results suggest that regardless of which shared socioeconomic pathway is realized, a significant portion of the North American boreal biome will continue to shift towards landscapes that feature woody vegetation

height increases. These pathway-agnostic results complement a study reporting a negative impact on growth and survival of tree species in the southern boreal with modest warming[60] and contribute to the ongoing discussion of the changes in North American boreal vegetation[44,61,62] with evidence that links the site history and current pattern of vegetation structure to the spatial patterns of its fate. The existing, climatically-driven forest gradient contextualizes how the structure of today's landscapes have both shifted in recent decades, and how they may shift in the future. They should

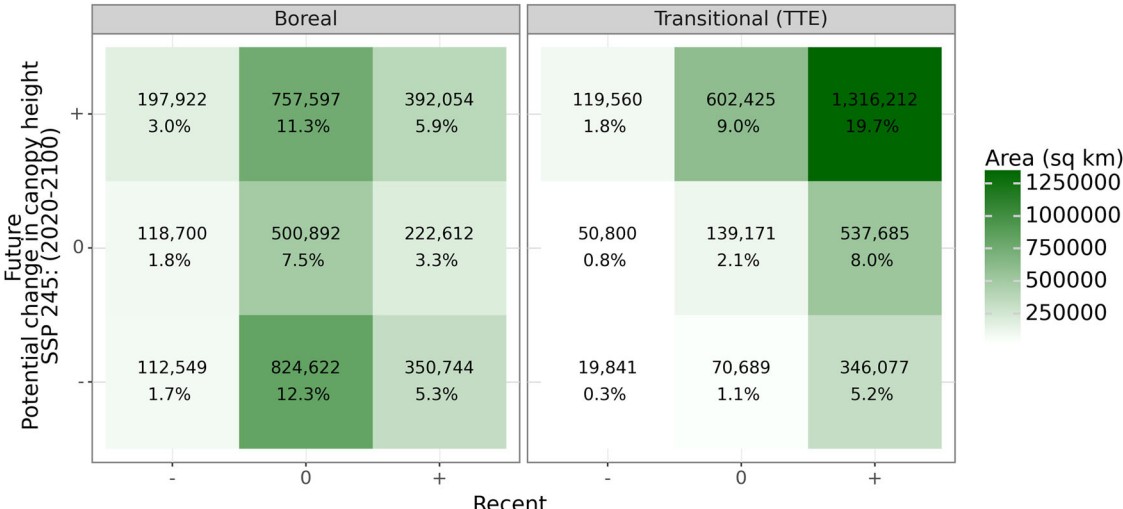

**Fig. 5 | A landscape-scale area summation and classification of the direction of structural changes from future (2020–2100) and recent (1984–2020) time periods across the North American boreal forest and transitional landscapes of the TTE.** Changes are classified based on median values captured across individual landscapes as negative (decreasing; -), no change (stable; 0), or positive (increasing; +).

Recent changes are from observations of 1984–2020 tree canopy cover trends and classes of future changes are from predictions of future (2100) potential canopy height changes, assuming SSP245, from current conditions. Numbers (top) represent the total area of each class and (bottom) the proportion of each class's area relative to the entire North American study domain.

be understood in light of studies reporting species-specific diversity in growth responses based on cold tolerance[63,64], and those considering the effects of disturbance agents on growth projections[65]. Interestingly, the fate of boreal forest structure through 2100 varies in strength spatially, but is similar across each of the 4 CMIP6 climate scenarios, providing evidence that an ongoing biome shift will continue. Furthermore, landscapes across the North American boreal and TTE for which predictions indicate the potential for widespread change in height suggest sites to target for detailed monitoring and prediction that account for local scale factors.

Across these landscapes, recent trends in forest structure may be indicative of future dynamics. If continuity in the direction of forest structure change persists through this century, then recent positive trends are evidence of conditions that may continue to support forest growth through 2100. These results reveal a markedly unevenly changing boreal forest domain, whose variation is associated with the spatial gradient in forest structure, in both the near future as well as across recent decades.

## Potential for a concentration of growth in transitional forests through 2100

Transitional landscapes that are currently associated with, or adjacent to, the boreal-tundra ecotone have the potential for a concentration of growth in canopy height while current interior boreal forest landscapes are projected to experience highly variable growth, regardless of which particular CMIP6 scenario is realized in the future. This general pattern is consistent with earlier retrospective assessments of forest productivity using spectral metrics and field measurements of tree secondary growth across bioclimatic gradients in boreal Alaska[66]. Predictions of potential height change show that the least forested (sparse forest and transitional) portions of the study domain feature the most positive changes in vegetation structure through 2100. These general patterns of change in the potential vegetation height through this century are consistently positive in non-forest and transitional (TTE) landscapes across all climate scenarios, differing most prominently in the magnitude of the positive change. The concentration of these positive potential changes in height in transitional forests suggest continued shifts in the vegetation structure of the North American boreal forest, resulting in an increase in the height of vegetation consistent with tall shrub and short-stature forest cohorts, as well as potentially a more gradual transition in vegetation structure from the current boreal forest poleward through to contemporary Arctic tundra domains. However, these predictions are based

on the current relationship of the vegetation gradient with climate, and it is possible that the strength of this response is non-stationary across the coming decades. Furthermore, the transitional landscapes discussed here do not include those along the southern warm edge of the boreal forest. Here, the magnitude, direction, and extent of a southern boreal shift may be responding to a different set of forces associated with a mix of management and drought-stress that reduce growth and increase mortality[60]. Therefore, the potential for a concentration of growth in transitional forests does not necessarily imply an increase in the overall extent of the boreal domain.

## Potential future change and recent trends suggest persistence of shift across transitional landscapes

Recent trends in forest structure in the boreal forests and associated transitional landscapes suggest that observed ongoing changes that are consistent with a biome shift have the potential to persist through 2100. This apparent persistent shift is quantified using median landscape-scale estimates of structure change, a way to generalize gridded results and report them in terms of coarser (mesoscale) landscape units (i.e., hydrobasins). With this approach, a shifting biome is one in which there is a clear directionality of change within specific landscapes, and within groups of landscapes that exhibit similar characteristics (in this study we use similarity in forest structure gradients within landscapes). In this case, we note that an apparent shift is occurring, and will continue to occur, because of both the similarities within landscape classes and of the consistent patterns of variation across different classes. With this, we can understand where site-level changes such as canopy infilling or height increases may generally take place, but cannot identify the specific sites on the landscapes that would show how these changes manifest. An examination that incorporates how seasonality, species type, and other functional characteristics, along with belowground, wildfire and permafrost dynamics, and hydrological changes will interact with structural changes may resolve finer-scale details of how these changes manifest at sites within landscapes. As such, our results cannot be used to quantify some specific types of shifts that are frequently of interest, such as a linear rate of change of a treeline advancing in a particular direction. Yet, the methods applied highlight transitional landscapes as hotspots of recent and predicted future positive vegetation structure change in the North American boreal forest.

This persistent shift, where there is alignment in the direction of recent and future changes in transitional extents, emerges from a variety of

independent spaceborne remote sensing observations. Both the Landsat multi-decadal record of boreal-calibrated tree canopy cover and the current ICESat-2 lidar observations of vegetation height are independent observations of vegetation structure. The model linking current lidar observations of vegetation height with bioclimatic variables together with recent trends in tree cover account for over a century (126 years) during which densifying canopy cover and increasing canopy height form the basis of this assessment of boreal forest structure change across transitional landscapes.

### Highly variable growth potential across interior boreal landscapes

Overall, landscapes that feature contemporary forest structure patterns consistent with the interior boreal show no strong change in median potential height change across all SSPs through 2100. These predictions suggest that, at landscape scales, any increases in growth will be offset by dieback and mortality. However, the variation associated with these near 0 median values suggest strong heterogeneity in these predicted changes in vegetation structure across the North American portion of the biome across this century. This is consistent with findings that report recent shifts in the boreal extent that are based on tree cover and NDVI change[1,2], where strong variation in recent tree cover changes create patterns of recent boreal change that are associated with post-disturbance recovery.

### Site factors and stochastic processes will modify predictions

The results on potential canopy height change are based solely on predicted changes in temperature and precipitation variables, a small, geographically coarse subset of the complete set of environmental drivers that operate across a range of scales to control vegetation canopy height. As such, the patterns of predicted height from our maps are not sensitive to site-scale factors of tree growth including the presence/absence of current seed sources, competition dynamics, composition, and germination potential. Predictions that indicate forest height growth where currently no forests exist suggest only the potential for growth in the height of woody structure, based on these climate variables, if other controlling factors such as adequate seed sources and soil conditions for germination and survival also support the potential for positive forest change.

These predictions also do not directly account for the variety of stochastic processes that alter vegetation in the high northern latitudes. The amount, distribution, size, and severity of disturbance factors from fire, harvest, permafrost degradation, and insect infestations will continue to influence forest growth, mortality, dieback, and competition through 2100. While some of these factors of site history do in fact manifest in the current heights of vegetation, this accounting of these controls of vegetation structure is indirect, and the effect of these factors on vegetation height is often non-linear[67]. The way vegetation-climate-fire feedback cycles will reinforce or de-stabilize current vegetation patterns are not handled in our models and increasing boreal fire occurrence and severity will likely alter the potential changes we report[68–70]. For example, shifting patterns of vegetation structure such as growth within the northern transitional forests may introduce new fire dynamics associated with novel fuel loads, however the manner in which they will positively influence a continued shift towards trees and shrub is uncertain[71,72]. In addition, while we explore potential change in vegetation height, we do not examine the concomitant potential change in vegetation type. Forest transitions from conifer to deciduous can affect the fire cycle, growth potential, surface energy balance, and carbon sequestration scenarios of the future[3,69,73,74]. These processes, in particular those associated with wildfire, are difficult to predict for specific pixels but be more easily anticipated for broader zones[75], will modify the potential changes we report. The contribution of these stochastic processes to non-stationary responses of boreal forest structure to climate should be explored on a consistent and regional basis in order to refine the potential for boreal forest structure change reported in this study[76,77].

## Methods

In this study, we use current vegetation structure patterns associated with a latitudinal forest gradient to link recent tree cover trends with future potential canopy height change. To do this, we identified a study domain across a latitudinal forest-tundra gradient in North America, compiled observations of current canopy height, assembled height predictor variables, built and applied a height prediction model used to derive ensemble predictions of potential canopy height for current and future time periods, assessed existing recent tree canopy cover trends, and summarized both predictions and recent trends at landscape scales using hydrological catchment basins.

### Identifying a study domain across a forest gradient

We identified a study domain that included the continental North American boreal forest[78] and the continuous, discontinuous, sporadic, and isolated permafrost extent[79]. This domain encompassed a gradient of current forest structure (Supplementary Fig. 1), area with no forest, and a gradient of current permafrost extent classes. Within this study domain, we developed a nested approach to training and testing, prediction, and analysis for model predictions of potential boreal forest canopy height for future time periods. Model training and testing was performed across the full domain, which included areas of non-forest primarily in the northernmost latitudes within continental North America as well as warm permafrost domains (primarily sporadic, isolated, and discontinuous) south and west of the North American boreal extent. Areas for which model predictions and analysis were performed excluded northernmost non-forest regions, focusing on the continental boreal extent and its surrounding 100 km buffered region, as well as boreal portions of the islands of eastern Canada (Supplementary Fig. 2).

### Compiling observations of current forest canopy height from spaceborne lidar

We compiled spaceborne lidar observations of current forest canopy height from ICESat-2 Land - Vegetation Along-Track Products (ATL08 v5) 20 m segment data. These data provide a broad sample of vegetation heights useful for sampling the broad domain of the boreal forest[57,80]. These observations were gathered from granules acquired seasonally during June-September in 2018–2022 for North America (45 N - 75 N, 50 W - 160 W). We used ATL08 data quality flags and values to limit the set of ATL08 observations to those most suited for analysis of boreal vegetation. As such, we filtered these observations to include only those associated with:

1. strong beams,
2. snow-free land surface,
3. cloud-free conditions,
4. night and low light (solar elevation angle <5 degrees),
5. a valid 98th percentile height range ≤30 meters & below ATL08 land-cover-specific height thresholds (Table S1),
6. a height difference from reference elevation <25 meters,
7. a total vertical geolocation error due to ranging and local slope <2.5 meters

Within the *training and testing* domain we created a 'canopy height flag' to identify and separate assumed erroneous reference ATL08 observations of canopy height. Using this flag, we set reference ATL08 observations to 0 m. This step was helpful in preparing a training dataset that included areas currently featuring no vegetation height, while also mitigating the bias in canopy heights present in filtered ATL08 data in the broad area of bare ground and tundra at the coldest northern edge of our study domain by reducing the overestimation of current and future vegetation structure in currently non-forested extents. To mitigate spurious observations at the high end, we also applied a height cap, by assigning a height of 30 m to those observations greater than this value. This yielded 19,875,592 point-based ICESat-2 ATL08 observations of canopy height across the full *training and testing* domain (Supplementary Fig. 3). These filtered and flagged reference canopy height

https://doi.org/10.1038/s43247-024-01454-z                                                                                   **Article**

observations, that are transects of points corresponding to 20 m segments along the tracks of strong beams that are nominally 11 m wide, represent the top of the vegetation canopy with the 98th percentile relative height (RH98). These point observations of RH98 were then gridded to 1 km, using a 'maximum' aggregation rule whereby the maximum RH98 value from all points within a 1 km cell became the cell's estimate of maximum canopy height. This generated a set of 1,082,746 ICESat-2 ATL08 gridded canopy height observations that approximates the tallest forests in each grid cell for circa 2020. A portion (80%) of these observations were used to train a model for predicting canopy height based on a set of predictor variables described below.

### Assembling predictor variables of potential forest canopy height

We assembled predictor variables from coarse-scale gridded (≥250 m) environmental covariates describing soil characteristics and climate conditions, and combined them with observations of current canopy height from ICESat-2 to model potential forest canopy height. Here, potential canopy height is defined as a metric that describes the maximum height of forest canopy that may be possible given the current relationship of canopy heights with these particular coarse-scale covariates, which are described below. They represent only a portion of the overall set of environmental covariates that determine the height of vegetation, and are used in this study to estimate even coarser (mesoscale) patterns of potential height. We modeled potential height for five time periods that include one current period (c. 2020) and four 20-year future time-periods (2021–2040, 2041–2060, 2061–2080, and 2081–2100) using coarse-scale environmental covariates from 2 groups.

The first group comprises 3 predictor variables that are representative of soil characteristics. These are gridded variables that estimate permafrost probability for 2000–2016[81], depth to bedrock, and organic carbon density at a depth of 0 m[82,83]. These variables provide only a rough representation of general characteristics associated with forest presence and growth, and are not intended to be an exhaustive set that accounts for the full variation of belowground site factors. Topography was used in the modeling process that estimated these characteristics and thus was not included separately. While these predictors vary spatially, they were held constant across the temporal domain of our study, in spite of the potential for change, particularly for permafrost. As such, they serve to impose some fundamental constraints to how vegetation structure changes through time, however because of their static nature their overall contribution to future vegetation is not fully developed in our modeling.

The second group comprises 19 predictor variables describing climate conditions across near current and future time periods that are provided as a suite of bioclimatic variables from the WorldClim Version2 dataset[84]. These variables, derived from monthly temperature and precipitation to describe the seasonality of environmental factors relevant to ecological modeling, were monthly averages over 20-year time-periods gridded at 30 s spatial resolution. For the 4 future time periods, we used time-period-specific predictions of those 19 bioclimatic variables from downscaled future predictions from Phase 6 of the Coupled Model Intercomparison Project (CMIP6)[85]. These bioclimatic variables were derived from 7–9 individual global climate models (GCMs) for 4 climate scenarios based on the CMIP6 Shared Socioeconomic Pathways (SSP) for a total 134 bioclimatic scenarios (Supplementary Fig. 4). The bioclimatic predictors from the climate projections of the 4 CMIP6 SSPs were used as a way to account for the resulting changes to the North American boreal ecosystem stemming from a range of possible changes to the climate system that are driven by emission and land use scenarios related to trajectories imposed by human development[59].

In total across these two groups, 22 predictor variables were used to model potential canopy height for one near current and four future time periods. These methods were designed to build predictions of the responses of current and future boreal forest structure to broad-scale environmental factors to examine the patterns in how vegetation structure may be affected by changes in climate.

### Building and applying a canopy height model

We built a model to predict canopy height with the grid-based ICESat-2 ATL08 lidar observations of canopy height circa 2020 and the corresponding set of 22 predictor variables. To generate a set of model training data, we built gridded stacks with the predictor variables and linked them to corresponding grid cells of mean canopy height from grid-based observations of height. Training data was gridded to 30 s (~1 km) grid cells across the extent. We then compiled a model prediction database by extracting the values of each of these 22 predictor variables (using the 30 s resolution version of the bioclimatic variables) to each gridded canopy height reference value derived from the 20 m point-based ICESat-2 ATL08 observations. We used this database for model training.

We trained a Random Forest regression model (using the cuML python package) on 80% of the grid cells, limiting the regression to 100 trees. This regression achieved an $R^2$ of 0.63 on the training data. Using permutation importance, the model ranked the most impactful variables for determining the variation in canopy height as annual mean temperature, precipitation of wettest quarter, and permafrost probability, with annual mean temperature the dominant factor associated with the variation in canopy height than any other variable in the model (Supplementary Fig. 5).

We then applied this model to map current canopy height and future potential canopy height. This model application procedure yielded 2.5 degree resolution grids of mean canopy height for current conditions as well as the potential mean canopy height for each individual GCM, across each SSP and future time period. Model performance was further evaluated by comparing the distributions of current canopy height observations with corresponding current canopy height predictions from the model.

### Deriving ensemble future canopy height and change predictions

We assembled the individual future potential canopy height predictions available from each GCM for each SSP and time period into ensemble predictions of future canopy height. To do this, we computed gridded maps of the median value of all individual GCM predictions for each SSP and future time period. This calculation operated pixel-wise, based on the set of individual canopy height predictions from the available SSP- and time-period-specific GCMs summarized in Supplementary Fig. 4. Then, we calculated maps of the differences between these future ensemble predictions and the current predictions. This step served to normalize the future predictions of potential canopy height with the same model-based predictions of current canopy height. This normalization was included to mitigate biases inherent in the model-based predictions of height. This produced 16 ensemble-based difference maps representing the potential differences in the future mean canopy height of vegetation for the 4 time-periods across the gradient of forest in our study domain according to the 4 CMIP6 SSPs.

### Assessing recent site history: multi-decadal tree cover change

We assessed recent site history for 30 m pixels across the study domain built from a multi-decadal time series of maps derived from Landsat-based (Landsat Thematic Mapper, Enhanced Thematic Mapper, and Operational Land Imager) boreal tree canopy cover estimates[86,87]. The time-series of boreal-calibrated tree canopy cover (TCC) maps exist for each year from 1984 ton2020 for the boreal forest biome[88], and are available through search.earthdata.nasa.gov. These TCC maps estimate the proportion of tree canopy >5 m in height for each 30 m observation. The yearly time step of these maps across 36 years provides a robust basis for estimating the trend in these TCC estimates. From this time series, the recent forest structure change was calculated per-pixel using ordinary least squares regression. These calculations returned a slope and corresponding p-value estimate per-pixel for all available 30 m gridded estimates of TCC. This map was compiled for the boreal forest extent, a subset of this study domain.

### Summarizing landscape-scale forest structure change

We assessed landscape (mesoscale) forest structure change patterns in both the predicted changes in forest height and recent site history in tree canopy cover across the boreal-tundra forest structure gradient. To do this, the study

domain was divided into level 8 (L8) hydrobasins[89]. These hydrobasins provide a means to summarize gridded data across multiple spatial scales in a geographically coherent manner by grouping observations according to similar catchment basins. Thus, they define the spatial bounds of the landscapes referenced in this study.

These landscapes were stratified into 6 classes based on forest structure pattern (the spatial gradient in current tree canopy cover)[22]. To achieve this classification, we used the set of landscapes associated with the 4 forest gradient classes in and near the taiga-tundra ecotone (TTE) from Montesano et al. 2020. Then, we appended to this set the remaining L8 hydrobasins that intersected our study domain and were associated with the boreal forest, which included a portions of the hydrobasins on the islands of eastern Canada (Anticosti, Prince Edward, Cape Breton, and Newfoundland). This appended set was divided into 2 classes according to the landscape's position relative to the classified TTE landscapes; '*Boreal Forest (taiga)*' (representative of interior boreal forest generally south of, and warmer than, TTE landscapes) or '*Non-forest (tundra)*' (generally north of, and colder than, TTE landscapes).

We used the spatial extent of each of these 16,559 landscapes to extract a statistical summary of potential future and recent forest structure change. This extraction procedure resulted in a database populated with the landscape median value of all gridded estimates of potential canopy height difference (current predicted height subtracted from future predicted height) for each of the 16 ensemble maps of future predictions of potential canopy height. Then, we performed this same extraction for the recent tree canopy cover trend map, returning the median trend (linear model slope) value for each landscape for which TCC trends had been calculated ($n = 12,734$). These median landscape values capture the central tendency of distributions composed of the gridded set of potential height differences or tree canopy cover trend for each landscape polygon. This procedure resulted in a database describing landscape scale patterns of potential future changes in canopy height and recent trends in tree canopy cover.

## Data availability

Data in this study include future climate gridded bioclimatic variables available from https://www.worldclim.org/data/cmip6/cmip6climate.html, permafrost probability from https://doi.pangaea.de/10.1594/PANGAEA.888600, SoilGrids data from www.soilgrids.org, and annual boreal tree canopy data from https://doi.org/10.3334/ORNLDAAC/2012. All other data and code relevant to the results, including the cleaned (filtered and clipped) ICESat-2 ATL08 Version 5 data used for model development, are accessible via: https://github.com/nasa-nccs-hpda/boreal_height_cmip6.

## Code availability

Code used to generate results is available via https://github.com/nasa-nccs-hpda/boreal_height_cmip6.

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

## Acknowledgements

This work was supported with grants from the NASA Terrestrial Ecology Program proposal #'s 21-TE21-0047 and 16-CARBON16-0124. Data processing for this study was performed on the EXPLORE system at the NASA Center for Climate Simulation and on the NASA Multi-Mission Algorithm and Analysis Platform. The authors acknowledge the contributions to dataset development and hosting provided by TerraPulse, Inc.

## Author contributions

This study was designed by P.M.M., M.C., and C.N. The manuscript was written by PMM and MF. Analyses and visualizations were performed by P.M.M., M.F., and J.L. Manuscript reviews were performed by MM, G.F., M.C., C.N., M.F., J.L., J.S., and P.M.M., P.M.M., M.F., J.L., and J.S. contributed to dataset development. Funding for this work was provided by G.F. and C.N.

## Competing interests

The authors declare no competing interests.
