## [Peer Review File · Communications Earth & Environment]

4th Mar 24

Dear Dr Montesano,

Your manuscript titled "Persistence of a North American boreal forest shift through 2100: transitional landscapes feature recent and predicted growth in vegetation structure" has now been seen by 2 reviewers, and we include their comments at the end of this message. They find your work of interest, but some important points are raised. We are interested in the possibility of publishing your study in *Communications Earth & Environment*, but would like to consider your responses to these concerns and assess a revised manuscript before we make a final decision on publication.

We therefore invite you to revise and resubmit your manuscript, along with a point-by-point response that takes into account the points raised. Please highlight all changes in the manuscript text file.

Please use the following link to submit your revised manuscript, point-by-point response to the referees' comments (which should be in a separate document to any cover letter), a tracked-changes version of the manuscript (as a PDF file) and the completed checklist:

[link redacted]

We hope to receive your revised paper within six weeks; please let us know if you aren't able to submit it within this time so that we can discuss how best to proceed. If we don't hear from you, and the revision process takes significantly longer, we may close your file. In this event, we will still be happy to reconsider your paper at a later date, as long as nothing similar has been accepted for publication at *Communications Earth & Environment* or published elsewhere in the meantime.

Please do not hesitate to contact us if you have any questions or would like to discuss these revisions further. We look forward to seeing the revised manuscript and thank you for the opportunity to review your work.

Best regards,

Alienor Lavergne, PhD
Associate Editor
Communications Earth & Environment

EDITORIAL POLICIES AND FORMATTING

We ask that you ensure your manuscript complies with our editorial policies. Please ensure that the following formatting requirements are met, and any checklist relevant to your research is completed

and uploaded as a Related Manuscript file type with the revised article.

Editorial Policy: Policy requirements (Download the link to your computer as a PDF.)

Furthermore, please align your manuscript with our format requirements, which are summarized on the following checklist:

Communications Earth & Environment formatting checklist

and also in our style and formatting guide Communications Earth & Environment formatting guide .

*** DATA: Communications Earth & Environment endorses the principles of the Enabling FAIR data project (<http://www.copdess.org/enabling-fair-data-project/>). We ask authors to make the data that support their conclusions available in permanent, publically accessible data repositories. (Please contact the editor if you are unable to make your data available).

All Communications Earth & Environment manuscripts must include a section titled "Data Availability" at the end of the Methods section or main text (if no Methods). More information on this policy, is available at <http://www.nature.com/authors/policies/data/data-availability-statements-data-citations.pdf>.

If a community resource is unavailable, data can be submitted to generalist repositories such as figshare or Dryad Digital Repository. Please provide a unique identifier for the data (for example a DOI or a permanent URL) in the data availability statement, if possible. If the repository does not provide identifiers, we encourage authors to supply the search terms that will return the data. For data that have been obtained from publically available sources, please provide a URL and the specific data product name in the data availability statement. Data with a DOI should be further cited in the methods reference section.

REVIEWER COMMENTS:

Reviewer #1 (Remarks to the Author):

COMMSENV-24-0402-T Review report

In this manuscript entitled "Persistence of a boreal forest shift in North America through 2100: transitional landscapes feature recent and predicted growth in vegetation structure regardless of climate scenario," the authors present an analysis of potential changes in tree and shrub cover through the end of the century. They propose a somewhat new modelling method in which the current gradient in canopy height is modelled as a function of 22 climate predictor variables. The model is then used to project future changes in canopy height with future climate predictions with a classic space-for-time approach. Such analyses are needed, given the important changes in forest ecosystems that have been observed in the last decades. Overall, the analyses conducted were sound and well-explained, and the whole manuscript was well-written and pleasant to read. For all these reasons, I think this manuscript is worth publishing in the *Communications Earth and Environment*. However, I have a few main comments that I'll develop below.

Although I like the effort to predict future changes with the space-for-time approach, it feels somewhat curious to present the predicted future trends before the observed recent trends. I understand that the model results presented are not based on the data presented in Figure 4 because the ICESat-2 lidar data are quite new and do not allow trend reconstructions in canopy height. However, I still suggest that the observed recent trends (Figure 4) should be presented before the projections of future changes.

My second concern is about the fact that historical factors are not considered in the model. Forest height is only modelled as a function of climatic variables, while historical factors such as fire are integral to the natural ecosystem dynamics (Gauthier et al. 2015). Some authors have developed similar approaches (though to a smaller extent) to model canopy height growth and accounting disturbance history (Pau et al. 2021, Danneyrolles et al. 2023). I realize that it would be rather complicated (though not impossible) to include the effect of fire in the model on this continental scale. I also note that the authors acknowledge in their last discussion section that "Site factors and stochastic processes will modify predictions." I suggest the authors elaborate further and more precisely on this aspect of the discussion. More particularly, they should discuss how the relation of vegetation height with climate will be non-linear due to fire feedback. Climate change is increasing fire in many parts of the North American boreal forests (Jain et al. 2017, Whitman et al. 2022, Barnes et al. 2023). Fire feedback includes short- to medium-term reductions in vegetation height, species change (i.e., changes in maximum potential height), and regeneration failure (Baltzer et al. 2021). Another aspect to discuss is that if tree or shrub vegetation moves north into the tundra, as predicted by the authors, the fire will also move north following the increase in fuel.

Reference

- Baltzer, J. L., N. J. Day, X. J. Walker, D. Greene, M. C. Mack, H. D. Alexander, D. Arseneault, J. Barnes, Y. Bergeron, Y. Boucher, L. Bourgeau-Chavez, C. D. Brown, S. Carrière, B. K. Howard, S. Gauthier, M.-A. Parisien, K. A. Reid, B. M. Rogers, C. Roland, L. Sirois, S. Stehn, D. K. Thompson, M. R. Turetsky, S. Veraverbeke, E. Whitman, J. Yang, and J. F. Johnstone. 2021. Increasing fire and the decline of fire adapted black spruce in the boreal forest. *Proceedings of the National Academy of Sciences* 118:e2024872118.
- Barnes, C., Y. Boulanger, T. Keeping, P. Gachon, N. Gillett, O. Haas, X. Wang, F. Roberge, S. Kew, D. Heinrich, R. Singh, M. Vahlberg, M. Van Aalst, F. Otto, J. Kimutai, J. Boucher, M. Kasoar, M. Zachariah, and F. Krikken. 2023. Climate change more than doubled the likelihood of extreme fire weather conditions in Eastern Canada. Imperial College London.

Danneyyrolles, V., Y. Boucher, R. Fournier, and O. Valeria. 2023. Positive effects of projected climate change on post-disturbance forest regrowth rates in northeastern North American boreal forests. *Environmental Research Letters* 18:024041.

Gauthier, S., P. Bernier, T. Kuuluvainen, A. Z. Shvidenko, and D. G. Schepaschenko. 2015. Boreal forest health and global change. *Science* 349:819–822.

Jain, P., X. Wang, and M. D. Flannigan. 2017. Trend analysis of fire season length and extreme fire weather in North America between 1979 and 2015. *International Journal of Wildland Fire* 26:1009–1020.

Pau, M., S. Gauthier, R. D. Chavardès, M. P. Girardin, W. Marchand, and Y. Bergeron. 2021. Site index as a predictor of the effect of climate warming on boreal tree growth. *Global Change Biology*:gcb.16030.

Whitman, E., S. A. Parks, L. M. Holsinger, and M.-A. Parisien. 2022. Climate-induced fire regime amplification in Alberta, Canada. *Environmental Research Letters* 17:055003.

Reviewer #2 (Remarks to the Author):

The study covers an important topic of interest to a large audience—spatial shifts in the North American boreal forest with climate change. This article specifically finds that boreal vegetation will expand into the tundra under a variety of future climate scenarios. This is indicated by projected increases in heights of vegetation with future projected warmer climates, in places where there are currently only scattered trees of short stature. Also, there will be variable changes in potential height within the current main range of the boreal biome, in some places trees are projected to become shorter as climate and site become unfavorable, while in other places tree heights will increase.

These findings are supported by the data and nicely illustrated in the figures. I think mapping current and potential future tree height is the best way to project the future condition of the boreal forest, given the recent availability of lidar data over large areas, and the historic usefulness of tree height as an indicator of forest condition.

I consider the findings to be novel because previously there were individual site studies in the field showing that trees are taking advantage of recent warming, or biome level analyses showing projected northward expansion of the boreal biome. However, this paper uses 30 m pixel data nested within 1 km grid squares, combined with models of tree response to climate projected into the future, to show how the whole North American boreal biome will change—in other words it has the maximum combination of spatial detail and large geographical extent that can be done at this time.

The data assembly processing and main conclusions are very good, based on the best available and most recent remotely sensed tree heights and data on site quality. I have used similar statistical methods in some of my published papers (e.g. random forests, regression trees using the bioclimatic variables), and it looks to me like the analyses here were done properly.

The Discussion also makes good points about the significant findings of the study and puts them in the context of previous literature. It also appropriately points out the limitations of the study with respect to disturbances and other factors that are not mapped that likely influence the current and

future height of boreal forests. However, using maximum potential height minimizes these influences by choosing grid cells with the 98th percentile in heights within a given 1 km grid square, since it is well known that even in recently burned areas, there are remnant stands of trees that represent to potential height of the forest.

I do recommend some minor revisions to make the paper suitable for publication:

The second sentence of the introduction is confusing. It makes it sound as if the entire boreal biome is experiencing an increase in presence of trees and shrubs, whereas it is the transition zone to tundra along the northern edge of the boreal biome that is experiencing increases in trees and shrubs.

The Discussion does not mention that along the southern ecotone of the boreal forest, there may be transitions to other biome types that exist just south of the boreal: grasslands (in central continental areas—the ‘prairie provinces’ of Canada) or to temperate forests (from eastern Minnesota across eastern Canada to the Atlantic Ocean). At a minimum, the Discussion should mention that these transitions were outside the scope of this study, but nevertheless are likely to occur, and perhaps that we don’t know the extent to which they will negate expansion in the north (i.e. whether the boreal as a whole will get smaller or larger).

With regard to the methods section in the supplementary material, everything in this paper hinges on modeling potential canopy height, and the explanation of this process needs to be totally clear. Am I correct in surmising that there were ca 1,100 30 x 30 m pixels in each square km, and that ca 22 of those 1,100 pixels with the tallest lidar-based tree height were averaged together and then used to represent each square km in the regression model? If so, I would state this explicitly either in/near lines 72-78 or lines 89-91 of the methods. Or if I am wrong state it correctly, or even give a formula if possible.

Finally, also in the supplementary/methods section, I noticed that Newfoundland and part of Nova Scotia are not included on the maps, whereas they are included on maps in the main article. A sentence or two about this discrepancy should probably be included in the methods, or else provide revised maps in the supplementary material.

Title

Persistence of a North American boreal forest shift through 2100: transitional landscapes feature recent and predicted growth in vegetation structure

Response to Referees

The Reviewers have provided a thorough critique of the manuscript and have shared a number of thoughtful comments (*italics*) and have provided responses to all of them (**bold**). We have carefully considered each comment and have, for nearly all comments, updated the manuscript in the manner that was suggested. We have otherwise provided justification for maintaining what we have. During this process, we have added a co-author who helped with dataset development and manuscript revision. The important updates made at the Reviewers' requests include:

1. Updated text in the Discussion on the importance of historical factors on potential forest height and to note we do not examine in detail the southern edge of the boreal.
2. Clarification of Methods on gridding of ICESat-2
3. An update to Supplementary Figure 2 to correct the omission of a small portion of the prediction and analysis domain in eastern Canada, and a corresponding update to the associated text.

The Reviewers' comments and the updates based on them have made this manuscript clearer and stronger, and we thank them for the consideration they have given to our work.

Sincerely,

Paul Montesano

Reviewer #1 (Remarks to the Author):

In this manuscript entitled "Persistence of a boreal forest shift in North America through 2100: transitional landscapes feature recent and predicted growth in vegetation structure regardless of climate scenario," the authors present an analysis of potential changes in tree and shrub cover through the end of the century. They propose a somewhat new modelling method in which the current gradient in canopy height is modelled as a function of 22 climate predictor variables. The model is then used to project future changes in canopy height with future climate predictions with a classic space-for-time approach. Such analyses are needed, given the important changes in forest ecosystems that have been observed in the last decades. Overall, the analyses conducted were sound and well-explained, and the whole manuscript was well-written and pleasant to read. For all these reasons, I think this manuscript is worth publishing in the Communications Earth and Environment. However, I have a few main comments that I'll

develop below.

Although I like the effort to predict future changes with the space-for-time approach, it feels somewhat curious to present the predicted future trends before the observed recent trends. I understand that the model results presented are not based on the data presented in Figure 4 because the ICESat-2 lidar data are quite new and do not allow trend reconstructions in canopy height. However, I still suggest that the observed recent trends (Figure 4) should be presented before the projections of future changes.

RESPONSE: The Reviewer's comment is understandable in that presenting the forest structure data in a chronological order is in some ways quite intuitive. Here, the focus is on predictions of potential future change, the predictive power of current structure, and the historical context that recent trends impart on these predictions. In large part because of our choice to drive our predictions of potential boreal forest change using various emissions scenarios, and then contextualize them with current spatial patterns of structure and recent temporal trends of structure. The authors assert that bringing in this historical context after the presentation of predictions and their variation across CMIP scenarios and geographic space, provide the most compelling explanation of the patterns in the predictions.

My second concern is about the fact that historical factors are not considered in the model. Forest height is only modelled as a function of climatic variables, while historical factors such as fire are integral to the natural ecosystem dynamics (Gauthier et al. 2015). Some authors have developed similar approaches (though to a smaller extent) to model canopy height growth and accounting disturbance history (Pau et al. 2021, Danneyrolles et al. 2023). I realize that it would be rather complicated (though not impossible) to include the effect of fire in the model on this continental scale. I also note that the authors acknowledge in their last discussion section that "Site factors and stochastic processes will modify predictions." I suggest the authors elaborate further and more precisely on this aspect of the discussion. More particularly, they should discuss how the relation of vegetation height with climate will be non-linear due to fire feedback. Climate change is increasing fire in many parts of the North American boreal forests (Jain et al. 2017, Whitman et al. 2022, Barnes et al. 2023). Fire feedback includes short- to medium-term reductions in vegetation height, species change (i.e., changes in maximum potential height), and regeneration failure (Baltzer et al. 2021). Another aspect to discuss is that if tree or shrub vegetation moves north into the tundra, as predicted by the authors, the fire will also move north following the increase in fuel.

RESPONSE: The Reviewer highlights the importance of historical factors and wildfire characteristics (frequency, severity, site responses, etc) in the prediction of future vegetation structure. In this work, such historical factors (eg., previous wildfire severity) are in some way accounted for insofar as they contribute to the expression of the current vegetation height that is captured with ICESat-2 data. However, this type of contribution of site history to predictions of vegetation height is coarse and indirect. We have elaborated on this missing control of height in our predictions at the Reviewer's request in the final paragraph of the Discussion. We point out how site-level fire dynamics may

alter predictions at the pixel-level significantly. Furthermore, we point out that other regional variability that we don't account for may continue to control patterns of vegetation structure change.

Reference

*Baltzer, J. L., N. J. Day, X. J. Walker, D. Greene, M. C. Mack, H. D. Alexander, D. Arseneault, J. Barnes, Y. Bergeron, Y. Boucher, L. Bourgeau-Chavez, C. D. Brown, S. Carrière, B. K. Howard, S. Gauthier, M.-A. Parisien, K. A. Reid, B. M. Rogers, C. Roland, L. Sirois, S. Stehn, D. K. Thompson, M. R. Turetsky, S. Veraverbeke, E. Whitman, J. Yang, and J. F. Johnstone. 2021. Increasing fire and the decline of fire adapted black spruce in the boreal forest. *Proceedings of the National Academy of Sciences* 118:e2024872118.*

Barnes, C., Y. Boulanger, T. Keeping, P. Gachon, N. Gillett, O. Haas, X. Wang, F. Roberge, S. Kew, D. Heinrich, R. Singh, M. Vahlberg, M. Van Aalst, F. Otto, J. Kimutai, J. Boucher, M. Kasoar, M. Zachariah, and F. Krikken. 2023. Climate change more than doubled the likelihood of extreme fire weather conditions in Eastern Canada. Imperial College London.

*Danneylolles, V., Y. Boucher, R. Fournier, and O. Valeria. 2023. Positive effects of projected climate change on post-disturbance forest regrowth rates in northeastern North American boreal forests. *Environmental Research Letters* 18:024041.*

*Gauthier, S., P. Bernier, T. Kuuluvainen, A. Z. Shvidenko, and D. G. Schepaschenko. 2015. Boreal forest health and global change. *Science* 349:819–822.*

*Jain, P., X. Wang, and M. D. Flannigan. 2017. Trend analysis of fire season length and extreme fire weather in North America between 1979 and 2015. *International Journal of Wildland Fire* 26:1009–1020.*

*Pau, M., S. Gauthier, R. D. Chavardès, M. P. Girardin, W. Marchand, and Y. Bergeron. 2021. Site index as a predictor of the effect of climate warming on boreal tree growth. *Global Change Biology:gcb*.16030.*

*Whitman, E., S. A. Parks, L. M. Holsinger, and M.-A. Parisien. 2022. Climate-induced fire regime amplification in Alberta, Canada. *Environmental Research Letters* 17:055003.*

Reviewer #2 (Remarks to the Author):

The study covers an important topic of interest to a large audience—spatial shifts in the North American boreal forest with climate change. This article specifically finds that boreal vegetation will expand into the tundra under a variety of future climate scenarios. This is indicated by projected increases in heights of vegetation with future projected warmer climates, in places where there are currently only scattered trees of short stature. Also, there will be variable changes in potential height within the current main range of the boreal biome, in some places trees are projected to become shorter as climate and site become unfavorable, while in other places tree heights will increase.

These findings are supported by the data and nicely illustrated in the figures. I think mapping current and potential future tree height is the best way to project the future condition of the

boreal forest, given the recent availability of lidar data over large areas, and the historic usefulness of tree height as an indicator of forest condition.

I consider the findings to be novel because previously there were individual site studies in the field showing that trees are taking advantage of recent warming, or biome level analyses showing projected northward expansion of the boreal biome. However, this paper uses 30 m pixel data nested within 1 km grid squares, combined with models of tree response to climate projected into the future, to show how the whole North American boreal biome will change—in other words it has the maximum combination of spatial detail and large geographical extent that can be done at this time.

The data assembly processing and main conclusions are very good, based on the best available and most recent remotely sensed tree heights and data on site quality. I have used similar statistical methods in some of my published papers (e.g. random forests, regression trees using the bioclimatic variables), and it looks to me like the analyses here were done properly.

The Discussion also makes good points about the significant findings of the study and puts them in the context of previous literature. It also appropriately points out the limitations of the study with respect to disturbances and other factors that are not mapped that likely influence the current and future height of boreal forests. However, using maximum potential height minimizes these influences by choosing grid cells with the 98th percentile in heights within a given 1 km grid square, since it is well known that even in recently burned areas, there are remnant stands of trees that represent to potential height of the forest.

I do recommend some minor revisions to make the paper suitable for publication:

The second sentence of the introduction is confusing. It makes it sound as if the entire boreal biome is experiencing an increase in presence of trees and shrubs, whereas it is the transition zone to tundra along the northern edge of the boreal biome that is experiencing increases in trees and shrubs.

RESPONSE: We agree with the Reviewer and have adjusted the wording accordingly.

The Discussion does not mention that along the southern ecotone of the boreal forest, there may be transitions to other biome types that exist just south of the boreal: grasslands (in central continental areas—the ‘prairie provinces’ of Canada) or to temperate forests (from eastern Minnesota across eastern Canada to the Atlantic Ocean). At a minimum, the Discussion should mention that these transitions were outside the scope of this study, but nevertheless are likely to occur, and perhaps that we don’t know the extent to which they will negate expansion in the north (i.e. whether the boreal as a whole will get smaller or larger).

RESPONSE: The Reviewer makes a good point that it may be important to be more explicit in that we are focusing our analysis of potential change in the transitional forests of the boreal-tundra, and not the southern limit of the boreal. We have updated the end of the 2nd paragraph of the Discussion accordingly.

With regard to the methods section in the supplementary material, everything in this paper hinges on modeling potential canopy height, and the explanation of this process needs to be totally clear. Am I correct in surmising that there were ca 1,100 30 x 30 m pixels in each square km, and that ca 22 of those 1,100 pixels with the tallest lidar-based tree height were averaged together and then used to represent each square km in the regression model? If so, I would state this explicitly either in/near lines 72-78 or lines 89-91 of the methods. Or if I am wrong state it correctly, or even give a formula if possible.

RESPONSE: We have added text at the Reviewer's request to clear up the misunderstanding associated with the preparation of ICESat-2 ATL08 canopy height data. There are no 30m pixels used in our ATL08 data processing. We start with ICESat-2 ATL08 observations (that are transects of points along the strong beam tracks) that represent, in this case, measurements of vegetation height along a 20m segment portion of each track (that is nominally 11m wide). The height metric widely used to represent the top of the canopy, and that we use here, is the 98th percentile relative height (RH98) measurement within each of these ATL08 observations. These point observations of RH98 are gridded into 1 km cells using a maximum rule (to represent the tallest lidar-based tree heights), resulting in transects of grid cells (following the ICESat-2 beams tracks) representing a maximum vegetation height. These grid cells are used to train and test the model.

Finally, also in the supplementary/methods section, I noticed that Newfoundland and part of Nova Scotia are not included on the maps, whereas they are included on maps in the main article. A sentence or two about this discrepancy should probably be included in the methods, or else provide revised maps in the supplementary material.

RESPONSE: - We thank the Reviewer for pointing this out. We have updated Supplementary Figure 2 showing that the prediction and analysis domains include the boreal hydrobasins found on the islands of eastern Canada (Anticosti, Prince Edward, Cape Bretton, and Newfoundland). We added text to lines 33-36 and 206-208 of the Methods to clarify this.

15th Apr 24

Dear Dr Montesano,

Your manuscript titled "Persistence of a North American boreal forest shift through 2100: transitional landscapes feature recent and predicted growth in vegetation structure" has now been seen by our reviewers, whose comments appear below. In light of their advice we are delighted to say that we are happy, in principle, to publish a suitably revised version in Communications Earth & Environment under the open access CC BY license (Creative Commons Attribution v4.0 International License).

We therefore invite you to edit your manuscript to comply with our format requirements and to maximise the accessibility and therefore the impact of your work.

EDITORIAL REQUESTS:

*****Please take care to match our formatting and policy requirements. We will check revised manuscript and return manuscripts that do not comply. Such requests will lead to delays. *****

SUBMISSION INFORMATION:

OPEN ACCESS:

Communications Earth & Environment is a fully open access journal. Articles are made freely accessible on publication under a CC BY license (Creative Commons Attribution 4.0 International License). This license allows maximum dissemination and re-use of open access materials and is preferred by many research funding bodies.

For further information about article processing charges, open access funding, and advice and support from Nature Research, please visit <https://www.nature.com/commsenv/article-processing-charges>

At acceptance, you will be provided with instructions for completing this CC BY license on behalf of all authors. This grants us the necessary permissions to publish your paper. Additionally, you will be

asked to declare that all required third party permissions have been obtained, and to provide billing information in order to pay the article-processing charge (APC).

[link redacted]

Best regards,

Joe Aslin

Deputy Editor,
Communications Earth & Environment
<https://www.nature.com/commsenv/>
Twitter: @CommsEarth

REVIEWERS' COMMENTS:

Reviewer #1 (Remarks to the Author):

The authors did an excellent job of considering my comments. I recommend publication and would like to congratulate the authors on their paper.

Reviewer #2 (Remarks to the Author):

The authors did an very good job with their response to my previous review. I do not have any further suggestions for revisions, and I think this revised version of the article is suitable for publication in Communications earth and environment.